# Efficient Antibacterial/Antifungal Activities: Synthesis, Molecular Docking, Molecular Dynamics, Pharmacokinetic, and Binding Free Energy of Galactopyranoside Derivatives

**DOI:** 10.3390/molecules28010219

**Published:** 2022-12-26

**Authors:** Faez Ahmmed, Anis Ul Islam, Yousef E. Mukhrish, Youness El Bakri, Sajjad Ahmad, Yasuhiro Ozeki, Sarkar M. A. Kawsar

**Affiliations:** 1Laboratory of Carbohydrate and Nucleoside Chemistry (LCNC), Department of Chemistry, Faculty of Science, University of Chittagong, Chittagong 4331, Bangladesh; 2Department of Chemistry, Faculty of Science, Jazan University, Jazan 45142, Saudi Arabia; 3Department of Theoretical and Applied Chemistry, South Ural State University, Lenin Prospect 76, 454080 Chelyabinsk, Russia; 4Department of Health and Biological Sciences, Abasyn University, Peshawar 25000, Pakistan; 5School of Sciences, Yokohama City University, 22-2, Seto, Yokohama 236-0027, Japan

**Keywords:** drug candidates, methyl β-d-galactopyranoside, antimicrobial, molecular docking, molecular dynamics, pharmacokinetic predictions

## Abstract

The chemistry and biochemistry of carbohydrate esters are essential parts of biochemical and medicinal research. A group of methyl β-d-galactopyranoside (β-MGP, **1**) derivatives was acylated with 3-bromobenzoyl chloride and 4-bromobenzoyl chloride in anhydrous *N*,*N*-dimethylformamide/triethylamine to obtain 6-*O*-substitution products, which were subsequently converted into 2,3,4-tri-*O*-acyl derivatives with different aliphatic and aromatic substituents. Spectroscopic and elemental data exploration of these derivatives confirmed their chemical structures. In vitro biological experiments against five bacteria and two fungi and the prediction of activity spectra for substances (PASS) revealed ascending antifungal and antibacterial activities compared with their antiviral activities. Minimum inhibitory concentration (MIC) and minimum bactericidal concentration (MBC) experiments were performed for two derivatives, **3** and **9,** based on their antibacterial activities. Most of these derivatives showed >780% inhibition of fungal mycelial growth. Density functional theory (DFT) was used to calculate the chemical descriptors and thermodynamic properties, whereas molecular docking was performed against antibacterial drug targets, including PDB: 4QDI, 5A5E, 7D27, 1ZJI, 3K8E, and 2MRW, and antifungal drug targets, such as PDB: 1EA1 and 1AI9, to identify potential drug candidates for microbial pathogens. A 100 ns molecular dynamics simulation study revealed stable conformation and binding patterns in a stimulating environment by their uniform RMSD, RMSF, SASA, H-bond, and RoG profiles. In silico pharmacokinetic and quantitative structure–activity relationship (QSAR) calculations (pIC_50_ values 3.67~8.15) suggested that all the designed β-MGP derivatives exhibited promising results due to their improved kinetic properties with low aquatic and non-aquatic toxicities. These biological, structure–activity relationship (SAR) [lauroyl-(CH_3_(CH_2_)_10_CO-) group was found to have potential], and in silico computational studies revealed that the newly synthesized MGP derivatives are potential antibacterial/antifungal candidates and can serve as therapeutic targets for human and plant pathogens.

## 1. Introduction

Microorganisms are responsible for a wide range of fatal diseases. The optimal approach to developing effective antimicrobial agents is to synthesize new chemicals and test their antimicrobial activities. Carbohydrates are organic substances that are the most abundant biomolecules on Earth, and they have a wide range of physical and physiological properties and several health benefits. The main function of carbohydrates is to provide energy. Carbohydrate-based compounds have come to the attention of researchers for making significant contributions to biological functions, such as cell development and cell proliferation, connections between several cells, and improvement of immune power [1]. Carbohydrates provide a major part of the energy that all organisms require for several biological works. Carbohydrates play a vital role in health and fitness, form a major portion of food, and assist greatly in growing body strength by producing energy. They are one of the three principal macronutrients that act as major energy providers; the other two are fats and proteins. For instant energy supply, sugars and starch work as fuel that helps one to perform physical activities perfectly. Carbohydrates add to the taste and appearance of a food item, thus making the dish tempting and mouthwatering. They also contribute to the metabolism process and inter-cell–cell interactions by supplying the required energy [2,3]. Another important characteristic of carbohydrate molecules is functioning as an anti-agent for several microbial organisms [4]. Compounds having aromaticity (aromatic and hetero-cyclic) were found to be enriched in biological capabilities in the literature review [5,6,7,8,9,10,11,12]. Generally, halogen, sulfur, and nitrogen-substituted aromatic molecules and their derivatives have strong potential to enhance antimicrobial efficacy [13,14,15,16,17,18]. Additionally, regioselective acylation and antimicrobial activity screening of carbohydrate compounds showed that the attachment of hetero-cyclic aromatic rings with electron-attracting or donating groups markedly increase the biological property of the precursor molecules [19,20,21]. Monosaccharide analogs are considered as having the potential for displaying broad-spectrum antimicrobial functioning in the case of both Gram-negative and -positive strains [18,19,20,21]. In the current investigation, the number of monosaccharide analogs were found to show notable inhibitory activity against cancer cells [22]. The attachment of aliphatic and aromatic groups to modify the hydroxyl group of the nucleoside and monosaccharide structure has been worthwhile in the development of potent antiviral [23,24,25,26,27,28] and antimicrobial candidates [28,29]. Keeping these features in mind, as well as the future target of searching for novel drug agents [30,31,32,33,34,35,36], in this investigation, we reported the biological screening of a number of β-MGP-based analogs **2**–**10** with some rarely used aliphatic and aromatic groups against seven pathogens, including molecular docking against the bacterial and fungal proteins. The docked complexes were tested to check their stability by a 100 ns molecular dynamics simulation. Furthermore, all the synthesized β-MGP analogs were investigated through density functional theory (DFT) optimization to explore their physicochemical and pharmacokinetic features.

## 2. Results

The research work reported here aimed to carry out regioselective bromobenzoylation of β-MGP (**1**) (Figure 1) and transformation of the synthesized 6-*O*-(4/3-bromobenzoyl) (**2**/**8**) into a number of substituted derivatives (Appendix A). This 6-*O*-(4/3-bromobenzoyl) (**2**/**8**) and its 2,3,4-tri-*O*-acyl derivatives (**2**–**10**) were employed as the test compound’s antimicrobial screening, along with the prediction of PASS and in silico studies.

### 2.1. Characterization

The methyl β-d-galactopyranoside (**1**) was initially converted to the 4-bromobenzoate **2** by treatment with 4-bromobenzoyl chloride in dry pyridine at −5 °C followed by the usual workup and purification and gave compound **2** in 70.55% as a crystalline solid. Recrystallization from EtOAc-*n*-C_6_H_14_ gave title (**2**) as needles, m.p. = 67–68 °C. The FTIR of this compound showed the following absorption bands: 1716 (C=O) and 3392~3497 cm^−1^ (br-OH) stretching. The formation of a single substitution product was clearly revealed by its ^1^H-NMR spectrum which showed one two-proton multiplet at δ 7.90 (as Ar-H) and one two-proton multiplet at δ 7.58 (as Ar-H), corresponding to the aromatic ring protons of one 4-bromobenzoyl group in the molecule. The downfield shift of C-6 to δ 4.85 (as dd, *J* = 11.0 and 6.3 Hz, 6a) and 4.63 (as dd, *J* = 11.1 and 6.2 Hz, 6b) from its usual value (~4.00 ppm) [19] indicated the attachment of the 4-bromobenzoyl group at position 6. The formation of 6-*O*-4-bromobenzoyl derivative (**2**) might be due to higher reactivity of the sterically less hindered primary hydroxyl group of the precursor molecule (**1**). The ^13^C-NMR spectrum also showed the presence of one 4-bromobenzoyl group by displaying the following expected resonance peaks: δ 165.20 (4-Br.C_6_H_4_CO-), 137.05, 133.28, 133.09, 129.69, 128.95, 126.08 (4-Br.C_6_H_4_CO-). The mass spectrum of compound (**2**) had a molecular ion peak at *m*/*z* 378.1402, corresponding to the molecular formula, C_14_H_17_O_7_Br. Complete analysis of the FTIR, ^1^H-NMR of this compound was in agreement with the structure accorded as methyl 6-*O*-(4-bromobenzoyl)-β-d-galactopyranoside (**2**) (Appendix A). Further support for the structure accorded to compound (**2**) was obtained by preparation of its lauroyl derivative (**3**) and myristoyl derivative (**4**). In the ^1^H-NMR spectrum of the compound **3**, the presence of three lauroyl groups in the molecule was ascertained by observing the following resonance peaks: δ 2.33 {6H, m, 3×CH_3_(CH_2_)_9_C*H*_2_CO-}, 1.63 {6H, m, 3×CH_3_(CH_2_)_8_C*H*_2_CH_2_CO-}, 1.28 {48H, m, 3×CH_3_(C*H*_2_)_8_CH_2_CH_2_CO-}, 0.88 {9H, m, 3×C*H*_3_(CH_2_)_10_CO-}. Its ^13^C-NMR spectrum also showed the presence of three lauroyl groups by displaying the following characteristic peaks: δ 172.50, 172.48, 172.46 {3×CH_3_(CH_2_)_10_CO-}, 34.38, 34.12 (×2), 31.90 (×3), 29.59 (×3), 29.45, 29.32 (×2), 29.24 (×3), 29.15, 25.01 (×2), 24.96, 22.67(×3), 22.65, 22.62 (×3), 21.72, 21.69, 20.09 (×2) {3×CH_3_(CH_2_)_10_CO-}, 13.51, 13.50, 13.48 {3×CH_3_(CH_2_)_10_CO-}. The molecular ion peak at *m/z* 924.9301 corresponding to the molecular formula, C_50_H_83_O_10_Br_,_ and the structure of the trilauroyate was ascertained as Methyl 6-*O*-(4-bromobenzoyl)-2,3,4-tri-*O*-lauroyl-β-d-galactopyranoside (**3**). Similarly, we were confidently able to propose a structure of the compound **4** as methyl 6-*O*-(4-bromobenzoyl)-2,3,4-tri-*O*-myristoyl-β-d-galactopyranoside (**4**) was ascertained by observing the following resonance peaks: δ 2.31 {6H, m, 3×CH_3_(CH_2_)_11_C*H*_2_CO-}, 1.60 {6H, m, 3×CH_3_(CH_2_)_10_C*H*_2_CH_2_CO-}, 1.25 {60H, m, 3×CH_3_(C*H*_2_)_10_CH_2_CH_2_CO-}, 0.87 {9H, m, 3×C*H*_3_(CH_2_)_12_CO-}. In order to prepare newer derivatives for biological evaluation, the 4-bromobenzoate (**2**) was also converted to the 3-chlorobenzote (**5**), 4-chlorobenzoate (**6**) and 4-*t*-benzoate (**7**). The structures of these derivatives were ascertained exclusively by analyzing their ^1^H-NMR, ^13^C-NMR, and mass spectra. Using the above procedure, compound **2** was converted to the 6-*O*-3-bromobenzoyl derivative (**8**). Treatment of compound **2** with 3-bromobenzoyl chloride in anhydrous pyridine yielded the compound (**8**) as needles. In its ^1^H-NMR spectrum, two one-proton doublets at δ 8.01 (as d, *J* = 7.1 Hz) and δ 7.22 (as d, *J* = 7.2 Hz), a one-proton singlet at δ 7.95 (Ar-H), and a one-proton triplet at δ 7.13 (*J* = 7.4 Hz, Ar-H) indicated the presence of one 3-bromobenzoyl group in the molecule. The ^13^C-NMR spectrum also supported the presence of one 3-bromobenzoyl group by displaying all the characteristic peaks. Compound **8** was also subjected to lauroylation and myristoylation to obtain compounds 2,3,4-tri-*O*-lauroate (**9**) and 2,3,4-tri-*O*-myristoate (**10**) in high yields. The structures of these derivatives were confidently confirmed by analyzing their ^1^H-NMR, ^13^C-NMR, and mass spectra (Appendix A). Thus, a number of derivatives of methyl β-d-galactopyranoside were prepared in good yields. All these newly synthesized products may be employed as important precursors for the modification of the β-d-galactopyranose molecule at different positions.

### 2.2. 2D-NMR Analysis

The signal from Ar-NH at the bottom left of the diagonal has a cross-peak labeled as Ar-NH, H-6b, connecting it to the signal from H-6b (Appendix A). Thus, the Ar-NH proton at approximately δ 7.90 is coupled to the hydrogen, whose signal appears at approximately δ 4.63 (i.e., H-6b proton). Similarly, the signal from H-6b is further connected by a cross-peak to the signal from H-5 to show the coupling between H-6b and H-5. The downfield shift of H-6a, H-1, H-2, and H-3 compared to the precursor analog (2) (Appendix A) demonstrated the attachment of 4-bromobenzoyl groups at C-6 positions of β-MGP. Assignments of the signals by analyzing their COSY, HSQC, and HMBC spectral experiments along with the ^13^C NMR spectrum confirmed the structure as methyl 6-*O*-(4-bromobenzoyl)-β-d-galactopyranoside (**2**).

### 2.3. Antibacterial Potentiality

Carbohydrate analogs were proved as antimicrobial agents by evaluating their biological capability via in vitro pathways [37]. All the designed compounds **2**–**10** were subjected to in vitro antimicrobial tests against several Gram-positive bacteria and Gram-negative strains. To determine MIC and MBC based on antimicrobial results, disk diffusion and broth microdilution procedures were employed, respectively. The outcomes of antibacterial evaluation are summarized in Table 1 and Table 2. Among all compounds, compound **9** displayed the maximum resistance with a notable zone of inhibition in the case of *B. cereus* (15.25 mm) and *B. subtilis* (13.00 mm), and compound **3** was also found to display a potential inhibitory property against *B. subtilis* (12.75 mm) and *B. cereus* (11.50 mm). These outcomes found a few fluctuating resistance zones for compounds **4**, **6**, and **8** which were active against only a single organism. Compounds **2**, **5**, **7**, and **10** were found to be inactive during the antibacterial test for both Gram-positive strains (Appendix A). Based on the data presented, derivatives **3**, **4,** and **6** exhibited a zone of inhibition against all three Gram-negative bacteria: *E. coli*, *S. typhi*, and *P. aeruginosa*, and derivatives **5** and **10** showed no inhibition against these. Based on the results for zones of inhibition, the compounds showed a higher activity against Gram-negative bacteria than against Gram-positive bacteria (Appendix A).

The antibacterial effects against the pathogens were determined by measuring the values of MIC and MBC of the most active β-MGP analogs. The results are listed in Appendix A and Figure 1 and Figure 2. The best antibacterial effects against the tested strains were observed for derivatives **3** and **9**, which showed MIC values in the range of 0.125–8.0 mg/L. Both the derivatives were active against all tested bacteria, and the best activity for these compounds was recorded against *B. subtilis* (0.125 mg/L). The lowest value of MBC was obtained for both derivatives (8.00 mg/L) against all pathogens. The highest value of MBC (16.00 mg/L) was obtained for derivative **9** against *B. subtilis*, *S. typhi*, and *P. aeruginosa*. The MBC values for these compounds for the other tested organisms are in the range 8–16 mg/L.

### 2.4. Antifungal Susceptibility

All the acyl β-MGP derivatives exhibited outstanding inhibition of the mycelial growth of both *A. niger* and *A. flavus* (Table 3). Derivatives **3** and **7**–**10** exhibited remarkable mycelial growth prevention against *A. niger* (78.81% ± 1.3%, 71.61% ± 1.3%, 72.88% ± 1.3%, 64.83% ± 1.3%, and 71.1% 9 ± 1.3%, respectively) and *A. flavus* (81.97% ± 1.3%, 78.69% ± 1.3%, 85.66% ± 1.3%, and 84.02% ± 1.3%, respectively) in the mycelial growth tests. Derivatives **2, 3,** and **5**–**10** were also effective against *A. niger* and *A. flavus*, and their zone of inhibition was higher than the standard antibiotic nystatin (Appendix A). Thus, the acylation of β-MGP improves antimicrobial activity.

### 2.5. SAR Analysis

The SAR study was performed for β-MGP derivatives based on their antimicrobial results, where compound **6** was found as the most active molecule for all the tested organisms. The SAR outcomes clarified that the insertion of several electron-enriched and electron-deficient groups firstly in the C-5 position and finally on the C-2, C-3, and C-4 positions of β-MGP structures markedly enhances the antimicrobial ability of the synthesized compounds. The external membrane of the Gram-negative organism consists of phospholipids, that have a pure hydrophobic property. Compound **6** was modified with a long hydrocarbon chain at the C-5 position which was also capable of hydrophobic interaction. As a result, hypothetically, it was suggested that compound **6** made hydrophobic interaction with the outer phospholipid membrane of bacteria. The 4-chlorobenzoyl benzoyl ring joined at the C-2, C-3, and C-4 positions and also followed the same procedure to show its antibacterial activity (Figure 3). Again, in the Gram-positive strain, compound **6** enters the bacterial membrane via a fatty peptidoglycan ledge (Figure 3). Here also may happen a hydrophobic interaction between the sugar environment of β-MGP analogs **6** and the peptidoglycan part of the cell wall. The rest of the inhibiting mechanism is often identified as being found in Gram-negative strains. The whole result suggested that this type of mechanistic approach may also be validated for compounds **3**, **8**, and **9**.

### 2.6. Predicted Antimicrobial Activities (PASS) Analysis

The antimicrobial spectrum was also predicted by applying the web server PASS to all the β-MGP derivatives **2**–**10**. The PASS results are expressed as Pa and Pi and are displayed in Table 4. For the β-MGP derivatives **2**–**10**, 0.32 < Pa < 0.58, 0.44 < Pa < 0.64, and 0.64 < Pa < 0.84 for antibacterial, antifungal, and antiviral activities, respectively. These results reveal that these molecules were more efficient against fungal and viral pathogens than against bacterial pathogens. The attachment of additional aliphatic acyl chains and the aromatic group increased antifungal activity (Pa = 0.645) of β-MGP (1, Pa = 0.342), whereas the insertion of lauroyl- and myristoyl-substituted groups slightly decreased the activity. However, derivative **6**, which has the 4-chlorobenzoyl-substituted aromatic group, exhibited the highest antifungal activity. The antiviral parameters of these derivatives were also predicted.

### 2.7. Thermodynamic Analysis

Free energy values of a molecule help determine the spontaneity of the reaction which reveals the stability of a product [38]. A higher (−)ve score is indicated to obtain thermodynamic stability. Compound **6** displayed the maximum free energy score (−6025.325 Hartree) as well as the maximum enthalpy score (−6025.324 Hartree) and maximum electronic score (−6025.445 Hartree) among all β-MGP analogs. A large value of dipole moment refers to the broadly polar nature of a molecule [39]. As summarized in Appendix A, compounds **2** and **5**–**8** have a fluctuating dipole moment score, which consequently affects the polar nature of a molecule, and modifies the binding energy, and binding mode of the ligand against the receptor protein. The value of dipole moments varied from a minimum of 3.321 Debye to a maximum value of 8.574 Debye. These results indicated that the change in hydroxyl (-OH) groups of β-MGP can finely raise the thermodynamic features of its analogs.

### 2.8. Frontier Molecular Orbitals (FMOs)

Molecular orbitals, one of the most fundamental factors towards the understanding of chemical reactivity and kinetic predictability, are known as frontier molecular orbitals [40]. The large energy difference of frontier molecular orbitals indicate strong chemical structural stability and weak reactivity. Generally, the outgoing of electrons from the stable level HOMO to the excited level LUMO demands additional energy. The HOMO and LUMO energies, HOMO-LUMO gap (∆), and all other chemical descriptors are shown in Table 5. Table 5 and Figure 4 combinedly confirm that compound **6** had a large energy difference (6.177 eV), and compound **2** had a comparatively smaller energy gap (5.263 eV) among all derivatives. Moreover, compound **4** was found to have chemical hardness and softness scores of 2.631 eV and 0.380 eV, respectively, where the hardness value was the maximum among all the derivatives.

### 2.9. Molecular Electrostatic Potential (MESP) Analysis

The molecular electrostatic potential (MESP) can identify ligands or protein binding areas and establish a suitable site for an electrophilic attack or nucleophilic attack. [41]. It may be used to determine how complete charges (both positive and negative) are scattered over the surface of an individual molecule [42]. The MESP of all β-MGP analogs was determined by geometry optimization using the B3LYP/3-21G basis set as shown in Figure 5. MESP is important as it can simultaneously show the molecular size and shape, as well as positive, negative, and neutral electrostatic potential regions in terms of color grading, which is crucial in the research of molecular structures, along with the physicochemical properties’ relationship [43]. MESP was calculated to forecast the reactive sites for the electrophilic and nucleophilic attack of the optimized structure of β-MGP (**1**) and derivatives **2**, **4**, **7,** and **8**. The red color displays the maximum negative area, which shows favorable sites for electrophilic attack, the blue color indicates the maximum positive area favorable for the nucleophilic attack, and the green color represents zero potential areas.

### 2.10. Molecular Docking Studies

The molecular docking technique is a powerful approach to predict the binding conformation of ligand molecules concerning a given biological macromolecule [44]. The method is also highly useful for determining the binding affinity of drug molecules to the biological target. In this study, all the synthesized compounds were docked with several antibacterial and antifungal targets. All derivatives showed the highest affinity for the MurF enzyme from antibacterial targets while lanosterol 14-*α*-demethylase for antifungal targets. The docking scores of the compounds are provided in Table 6. Derivative **9** was investigated as the best binder for both mentioned targets. In the case of MurF, derivative **9** docked at the center of three functional domains including the central domain, *N*-terminal domain, and C-terminal domain. It formed various van der Waals interactions with residues of all domains. It produced several alkyl interactions and pi-cations contacts. Details of the binding mode and interactions are presented in Figure 6. For the lanosterol14-*α*-demethylase target, the same compound accommodates itself in the functional pocket and interacts with several hydrophilic and hydrophobic residues. The central moiety of the compound was revealed to engage in most of the interactions (Figure 7). A current study reveals that β-MGP analogs strongly interact with receptors of *Escherichia coli*, *Bacillus subtilis*, and *Aspergillus flavus* with high binding scores [34]. Most of the compounds displayed π–π interactions with Phe399, Leu149, Arg381, and Trp382, along with Leu105, indicating strong binding with the active site. Phenylalanine is thought to be the prime site of pi–pi stacked and pi–pi T-shaped interactions caused the accessibility of small molecules to the active site [32,33]. The synthesized compounds are found to be bound within some catalytic residues such as Arg64, Arg95, Arg138, Arg146, Gln322, Cys129, Asn102, His101, Met110, and Met459 of the target proteins, which are responsible for several functions of these microorganisms.

### 2.11. Molecular Dynamics Simulations

The selected complexes were further subjected to molecular dynamics simulations to determine the dynamics of the complex and binding stability of derivative **9** with the antibacterial and antifungal receptors. To examine the structure dynamics stability of the complex, root means square deviations (RMSDs) of both receptors and derivative **6** were calculated throughout 100 ns. The root means square deviation of the C-alpha atoms found during dynamics was explored to realize the stability and the structural difference [32,33,34]. Figure 8 reveals that the analyzed analogs had the initial upper trend causing the flexible nature of the protein–ligand complexes at the starting stage. As a result, the protein–ligand complexes gained the steady-phase after 25 ns and continued to gain stability until the last segment of the simulations. Compound **9** showed reasonably higher RMSD than the other docked complexes, revealing the promising flexible property of all the complexes than other analogs. Therefore, all the protein–ligand complexes had RMSD less than 2.0 during the dynamics performance, resulting in the complexes’ stability. Both systems had noticeably good overall structural stability as no major deviations were pointed out. Some small RMSD jumps can be observed; however, these changes do not alter compounds binding to the receptors. These changes are, in fact, due to the large number of loops that allow flexibility of the receptors. The mean RMSD value of MurF-derivative **9** and lanosterol 14-*α*-demethylase-derivative **9** is 1.98 Å and 2.64 Å, respectively. Similarly, the RMSF analysis indicated the major receptors’ residues were stable in the presence of derivative **9**. This demonstrates favorable binding of the compound to the receptors, which allows for a stable compound binding mode to the receptors. The mean RMSF value of MurF-derivative **9** and lanosterol 14-*α*-demethylase-derivative **9** is 1.02 Å and 2.2 Å, respectively. The RMSD and RMSF plots of the systems are presented in Figure 8. Further, it is demonstrated that the strong binding of derivative **9** with the receptor is attributed to the continuous formation of hydrogen bonds between the compound’s chemical moieties receptor and active site residues. In Figure 9, the derivative interacts with the receptor’s residues with at least two hydrogen bonds along the length of simulation time. The analysis of the complex stability of systems was also conducted by calculating the radius of gyration (R_g_) and solvent-accessible surface area (SASA). The radius of gyration is the mass-weighted root mean square distance of the collection of atoms from their common center of mass. R_g_ is one of the important parameters to analyze the stability of proteins using MD simulation data. Figure 10A indicates that all complexes had an identical Rg score; again, lower degrees of deviation were found for the four simulated complexes. The solvent-accessible surface area of the simulations systems was explored to understand the changes in the protein surface area where the higher SASA (Figure 10B) defines the expansion of the surface area. In contrast, the lower SASA is related to the truncated nature of the complexes. The root means square fluctuations define the flexible nature of the complexes across the amino acid residues. Figure 10C reveals the stable nature of the complex of compound **9**.

### 2.12. Binding Free Energy Analyses

The trajectories of simulations were further analyzed in MM/GBSA and MM/PBSA binding free energies analysis to confirm the strong affinity of derivative **4** for the targeted receptor molecule. The contribution of each energy parameter to the net binding of the complexes is presented in Table 7. In both methods, the net binding energy is very stable for both complexes and reflected in the high intermolecular stable conformation and interaction profiles. In the case of MM/GBSA, the total binding energy of MurF-derivative **9** complex is −25.89 kcal/mol, while for lanosterol 14-*α*-demethylase-derivative **9**, the net binding energy is −33.27 kcal/mol. For both complexes, the electrostatic and van der Waals energies played significant roles; however, the solvation energy depicted a non-favorable contribution. In the case of MM/PBSA, similar to MM/GBSA, the net binding energy was −20.84 kcal/mol for MurF-derivative **9**, while it was −27.94 kcal/mol for lanosterol 14-*α*-demethylase-derivative **9**. The van der Waals energy to this total energy is very high compared to the electrostatic energy. Similarly, as observed in MM/GBSA, the solvation energy is negative in complex formation. However, in both MM/GBSA and MM/PBSA, the role of total non-polar solvation energy is positive in stable complex formation.

### 2.13. Pharmacokinetic Profile and Drug-Likeness Analyses

By considering the physicochemical and ADMET profiling data, compound **9** showed maximum clearance (1.663) and water solubility values (−3.196). Compound **7** showed maximum human intestinal absorption and skin permeability with 95.724 and 2.748, respectively. Maximum permeation of caco-2 cells and plasma protein binding showed by molecules **5** and **6** with values of 0.652 and 0.501, respectively. Maximum blood–brain barrier molecule **5** with 0.451 and distribution volume (Vd) showed by compound **6** and 0.052, respectively. Molecule **8** showed maximum logPS (the central nervous system (CNS) permeability) value of −2.314 (Table 8 and Table 9). Physicochemical and ADMET data revealed that most of the molecules maintained the drug-likeness rule. Compounds **2**–**5** and **8**–**10** fulfilled all the criteria of physicochemical and ADMET data.

Appendix A exhibits that most of the β-MGP species abstain from inhibiting all the enzymes, without CYP3A4 in the case of compounds **3** and **9**, so it may be suggested that the other compounds in the body system be metabolized by the P450 enzyme. The toxicity results of the β-MGP derivatives are described in Appendix A, and their high LD50 values (1.074–2.841) suggest that the analogs are lethal only at very high doses. A negative result in the AMES test suggests that an ester is not mutagenic. The results also suggest that all derivatives tested may not inhibit the hERG channel and may not have skin sensitivity.

### 2.14. Calculation of QSAR and pIC_50_

QSAR is a computational modelling approach for revealing correlations among the structural characteristics of chemical substances and biological activities. To calculate the QSAR and pIC_50_ score, multiple linear regression (MLR) equations were employed [45]. Our investigated compounds were found to fulfil all the required conditions and several QSAR and pIC_50_. The pIC_50_ values ranged from a minimum value of 3.67 to a maximum value of 8.15 (Table 10).

## 3. Materials and Methods

### 3.1. General Information

All the chemicals and solvents used in the synthesis and analytical work were Analar grade obtained from Sigma-Aldrich and Merck of Germany. Infrared spectra of the compounds were recorded on a calibrated Fourier-transform infrared (FTIR) spectrophotometer (IR Prestige-21, Shimadzu, Kyoto, Japan) using KBr pallets. ^1^H-NMR and ^13^C-NMR spectra were determined using Brucker DPX 400 MHz at WMSRC, JU, Bangladesh.

### 3.2. Synthesis

A suspension of methyl β-d-galactopyranoside (β-MGP, **1**) (100 mg, 0.515 mmol) was made, comprising dry *N*,*N*-dimethylformamide (3 mL) and triethylamine (0.15 mL), in a round bottom flask. It was cooled to −5 °C in an ice bath, and then, 4-bromobenzoyl chloride (121.8 mg, 1.1 molar eq.) was added. The reaction mixture was continuously stirred at the same temperature for 6 h, and then it was left standing overnight at room temperature (RT) with continuous stirring. The progress of the reaction was monitored by TLC (CH_3_OH-CHCl_3_, 1:6), which indicated full conversion of the starting material into a single product (*R_f_* = 0.52). A few pieces of ice were added to the flask and evaporated off. The resulting syrup was passed through a silica gel column and eluted with CH_3_OH-CHCl_3_ (1:6), providing the 4-bromobenzoyl derivative **2** (76.8 mg, 79.55%) as a crystalline solid. Recrystallization from EtOAc-*n*-C_6_H_14_ gave title (**2**) as needles. (Characterization of the spectral values are in the SI).

### 3.3. General Procedure for the Preparation of Lauroy Derivatives **3**–**7**

A solution of 4-bromobenzoyl derivative **2** (113.5 mg, 0.30 mmol) in dry *N*,*N*-dimethylformamide (DMF) (3 mL) and triethylamine (0.15 mL) was cooled to 0 °C and lauroyl chloride (0.34 mL, 5.0 molar eq.) was added. The reaction mixture was continuously stirred at the same temperature for 6 h, and then the reaction mixture was left standing overnight at room temperature with continuous stirring. The progress of the reaction was monitored by TLC (CH_3_OH-CHCl_3_, 1:6), which indicated full conversion of the starting material into a single product (*R_f_* = 0.52). The resulting syrup was purified by column chromatography (with CH_3_OH-CHCl_3_, 1:6, as the eluent) to afford the lauroyl derivative **3** (205 mg, 73.71%) as a white-colored solid. Recrystallization from EtOAc-*n*-C_6_H_14_ gave title (**3**) as needles.

Similar reaction and purification procedure was applied to prepare compound **4** as crystalline solid, compound **5** as needles, compound **6** as crystalline solid, compound **7** as needles, compound **8** as crystalline solid, compound **9** as needles, and compound **10** as needles. (Characterization of the spectral values are in the SI).

### 3.4. Biological Assessment

The antimicrobial assay of the compounds was conducted in the Department of Microbiology, University of Chittagong, Chattogram, Bangladesh. The test microorganisms (bacteria and fungi) (Appendix A) were collected from this Department.

#### 3.4.1. Disc Diffusion Test to Check Antibacterial Susceptibility

The in vitro antibacterial spectrum of the synthesized compounds was indicated by disc diffusion method according to CLSI protocol [46]. Antibacterial activities were indicated by a clear zone of growth inhibition around the disc. The inhibition zones were monitored after 24 to 48 h.

#### 3.4.2. Determination of MIC and MBC Using the Micro-Broth Dilution Method

MIC and MBC test of two compounds (**3** and **9**) against five bacteria have been investigated by disc diffusion microdilution methods. The actualization of a red color designated the growth of bacteria, and MIC and MBC were explained conspicuously.

#### 3.4.3. Screening of Mycelial Growth

In vitro antifungal activity of the synthesized (β-MGP, **1**) derivatives was evaluated by the ‘poisoned food technique’ [47] using a PDA medium. The diameter of radial growth of the test fungi was measured after 3 to 5 days. The experiment was repeated in triplicate.

### 3.5. Structure–Activity Relationship (SAR) Analysis

To identify the active portion of the synthesized molecule, a structure–activity relationship (SAR) study was performed. This well-known technology is often used in drug-designing processes to guide the acquisition or synthesis of new compounds with desirable properties. In the present study, the SAR study was analyzed according to the Kim [48] and Hunt [49] membrane permeation concept. The hydrophobicity of a compound is an important parameter with respect to such bioactivity as toxicity or alteration of membrane integrity because it is directly related to membrane permeation. Hunt [48] proposed that the potency of aliphatic alcohols is directly related to their lipid solubility through hydrophobic interaction between alkyl chains from alcohol and the lipid region in the membrane. Hydrophobic interaction might occur between acyl chains of (β-MGP, **1**) accumulated in the lipid-like nature of bacteria membranes. As a consequence of their hydrophobic interaction, bacteria lose their membrane permeability, ultimately causing the death of the bacteria.

### 3.6. PASS Enumeration

The online web application PASS (http://www.pharmaexpert.ru/passonline/, accessed on 12 April 2022) was used for determining the probable biological spectrum of the designed compounds [38]. Initially, the structures of β-MGP molecules were converted to SMILES formats by employing the SwissADME online tool (http://www.swissadme.ch, accessed on 12 April 2022). Then, the SMILES were input to the PASS web server to calculate the biological spectrum. PASS data are presented by Pa (probability for active molecule) and Pi (probability for inactive molecule). The acceptable score of Pa and Pi fluctuates in the range of 0.00–1.00, and usually, Pa + Pi ≠ 1, as these potentialities are predicted freely. Biological activities with Pa > Pi are only considered probable for select drug molecules.

### 3.7. Computational Details

#### 3.7.1. Geometry Optimization

In computational chemistry, quantum mechanical methods are widely used to calculate thermal, molecular orbital, and molecular electrostatic properties [50]. Geometry optimization and further modification of all synthesized analogs were carried out using the Gaussian 09 program [51]. DFT (3-21G) with Becke’s along with Lee, Yang, and Parr’s (LYP) (B) functional [52,53] were used to perform optimization. Water was used as a solvent environment during optimization. Then, Parr and Pearson’s interpretation of DFT and Koopman’s theorem [54] employed all the chemical descriptors considering the following formula.
$$\mathrm{Gap}\;(\Delta_{\varepsilon })=\varepsilon \mathrm{LUMO}-\varepsilon \mathrm{HOMO}$$
(1)$$\eta =\frac{\left[\varepsilon \mathrm{LUMO}-\varepsilon \mathrm{HOMO}\right]}{2}$$
(2)$$S=\frac{1}{\eta }$$
(3)$$\mu =\frac{\left[\varepsilon \mathrm{LUMO}\;+\;\varepsilon \mathrm{HOMO}\right]}{2}$$
(4)$$\chi =-\frac{\left[\varepsilon \mathrm{LUMO}\;+\;\varepsilon \mathrm{HOMO}\right]}{2}$$
(5)$$\omega =\frac{\mu^{2}}{2\eta }$$

#### 3.7.2. Preparation of Protein and Molecular Docking

Molecular docking studies were undertaken to determine the binding energies of the compounds for several antibacterial and antifungal targets; the solutions with the highest scores were used for molecular dynamics simulations. For validation purposes, co-crystallized ligands were used in each case and docked to the same position as reported in the crystal structure. In the case of antibacterial drug targets, the PDBs used were 4QDI, 5A5E, 7D27, 1ZJI, 3K8E, and 2MRW, and the antifungals used were 1EA1 and 1AI9. The docking procedure was executed using AutoDock 4.2.6 [55]. Before docking, both the proteins and compounds were processed in UCSF Chimera [56] for removing ligands that are not functionally relevant. The proteins/enzymes were minimized for energy using the steepest descent and conjugate gradient steps for 1000 steps in each algorithm. For docking purposes, the blind docking approach was utilized to provide the surface of a complete receptor for binding. This allowed us to determine which pocket is the best binding affinity area of the compounds. During the preparation of receptors, polar atoms were added, and partial charges were assigned. The Lamarckian genetic algorithm [57] was employed during docking studies, and for each compound, approximately 100 conformations were generated. To cover the active site along with the essential residues within the binding pocket, the three-dimensional grid box for docking simulation in which the box with the size of 45.338 × 66.087 × 55.237 was centered using the following dimensions, −1.513 × 3.784 × 6.897. The lowest binding energy docked conformation of each compound was selected as the best binder.

#### 3.7.3. Molecular Dynamics Simulation

The physical behavior and dynamics of the best energy docked complex from each antibacterial and antifungal docking were understood using all atoms’ molecular dynamics simulations performed for 100 ns using the AMBER20 simulation package [58]. Ligand parameters were generated through the general amber force field (GAFF) [59], while protein topologies were created through FF14SB [60]. The docked complexes were then submerged into a TIP3P water box where counter ions were added to obtain neutralized systems. Energy minimization was performed via 1500 steps of steepest descent and conjugate gradient algorithms. The systems were then heated to slowly increase the temperature to 36.85 °C and maintained through the Langevin algorithm. This was followed by restrained NVT ensemble equilibration and constant NPT ensemble for 1 ns equilibration. The long-range electrostatic interactions were calculated by the particle mesh Ewald method. Production simulation runs were performed for 100 ns under 36.85 °C and 1 bar. CPPTRAJ [61] was considered for structure stability analysis, and XMGrace was used for plotting [62].

#### 3.7.4. Calculation of Binding Free Energy

The MMPBSA [63] of AMBER was run on the simulation trajectories to estimate the binding free energies of the systems. In total, 100 frames were investigated, which were picked at regular intervals. Both Molecular Mechanics/Generalized Born Surface Area (MM/GBSA) and Molecular Mechanics/Poisson–Boltzmann Surface Area (MM/PBSA) methods were used for calculating net binding energies [64]. Details of the methods and procedure used were adapted from [65].

### 3.8. Pharmacokinetic and Drug-Likeness Prediction

Water solubility (log mol/L), log PAPP (caco-2 cell permeability), skin permeability, human intestinal absorption, plasma protein binding, blood–brain barrier permeation, volume of distribution, the central nervous system (CNS) permeability, renal OCT2 substrate, total clearance, CYP450 1A2 inhibitor, CYP450 3A4 inhibitor, CYP450 2C9 inhibitor, CYP450 2D6 inhibitor, Herg1 inhibition, human hepatotoxicity, AMES toxicity, skin sensitivity, LD50, of all the synthesized compounds were determined by using pkCSM [66].

## 4. Conclusions and Future Perspectives

The present study deals with the synthesis and in vitro and in silico analyses of a number of β-MGP-based molecules to explore their antimicrobial, thermodynamic, molecular docking, molecular dynamics, and drug-likeness properties. An initial biological test revealed that the insertion of various aliphatic chains and aromatic rings into the β-MGP structure can precisely improve the biological activities of the compounds. The outcomes were rationalized by subjecting molecular docking, which clarified that β-MGP derivatives **2**–**10** have promising binding interactions and binding energy with selected bacterial and fungal proteins. This result was further justified by molecular dynamic studies up to 100 ns, keeping in mind protein, which confirms the binding stability of the docked complex in the trajectory analysis (RMSD, RMSF, SASA, H-bond, and RoG profiles) and it means that the protein–ligand complex is highly stable in biological systems. Finally, these derivatives were investigated for their pharmacokinetic and QSAR properties, which stated that a combination of toxicity prediction, in silico ADMET prediction, and drug-likeness has promising results and most of the molecules maintained all drug-likeness rules as well as provided an interesting result in terms of biological activity. As this study was carried out using synthetic, antimicrobial, and in silico computational methods, ensuring these results would require further wet-lab experiments to be carried out under in vivo and in vitro conditions.

## Data Availability

Data is available in this article and Appendix A.

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
