# Peer review of "Efficient Antibacterial/Antifungal Activities: Synthesis, Molecular Docking, Molecular Dynamics, Pharmacokinetic, and Binding Free Energy of Galactopyranoside Derivatives"

_molecules, 2022, doi:10.3390/molecules28010219_

Round 1
Reviewer 1 Report
Dear
I believe this paper has merit to be published in the journal.
Regards
Author Response
Response to reviewer comments on Molecules 1978757
[N.B.: All corrections have been made by the blue color in the text and highlighted with yellow color]
Response to reviewer- 1 comments
Thank you so much for your favorable comments on the publication.
Kind regards
Reviewer 2 Report
I have reviewed your manuscript, and I am expressing my positive feedback. Your study is interesting for the readers of the Molecules journal, and the obtained results are promising. Since there is some space for improvements, I am requesting minor revisions according to the attached comments

Author Response
Response to reviewer comments on Molecules 1978757
[N.B.: All corrections have been made by the blue color in the text and highlighted with yellow color]
Response to reviewer- 2 comments
Comment-1: Computational details should be placed in a chapter before discussing the results.
Response: Thank you very much for your comments! According to the reviewer’s suggestion we have placed computational details in a chapter i.e. section 3.7. But we couldn’t place it before the result discussion because according to the journal style ‘Experimental section’ should be placed after the result discussion.
Comment-2: Please explain in detail the methodology adopted to study the structure-activity relationship (SAR) of the synthetized molecules.
Response: Correction has been done. We have explained the methodology of the structure-activity relationship (SAR) in detail (section 3.7).
Comment-3: “ω = μ2/2η ” should be corrected as follows: ω = μ 2/2η
Response: Correction has been done.
Comment-4: There is no statement in the article about at which phase the DFT calculations were performed. At gas phase? If so, since these molecules are designed as potential drug molecules and will be used in living organisms, wouldn't it be more appropriate to perform the calculations in water?
Response: Thank you ver much! Corrections have been done. We have performed the optimization using water as a solvent and according to your suggestion, we added the statement in the article (section 3.7.1).
Comment-5: In molecular docking calculations, why were the targets receptors (4QDI, 5A5E, 7D27, 1ZJI, 3K8E, 2MRW, 1EA1 and 1AI9) chosen? Has it been determined or known that antimicrobial activity has a mechanism based on interaction with these receptors? Perhaps the mechanism of action of the observed antimicrobial activity is through another target receptor. How can we be sure that it happens through these receptors?
Response: We have selected the target receptors based on our antimicrobial study. We performed our biological experiment against five bacteria (B. cereus, B. subtilis E. coli, S. typhi, and P. aeruginosa) and two fungi (A. niger and A. flavus) and found promising results. To rationalize the antimicrobial results and maintain the alignment of the study, we have chosen the receptors of the same organism.
Meanwhile, carbohydrate derivatives have promising antimicrobial properties, binding energy, and binding interaction against receptors of these organisms. Please have a look at the following articles in support of the evidence. Thank you!
- Amin, M.R.; Yasmin, F.; Hosen, M.A.; Dey, S.; Mahmud, S.; Saleh, M.A.; Hasan, ; Fujii, Y.; Yamada, M.; Ozeki, Y.; Kawsar, S.M.A. Synthesis, antimicrobial, anticancer, PASS, molecular docking, molecular dynamic simulations and pharmacokinetic predictions of some methyl β-D-galactopyranoside analogs. Molecules 2021, 26, 1–25.
- Amin, M.R.; Yasmin, F.; Dey, S.; Mahmud, S.; Saleh, M.A.; Emran, T.B.; Hasan, I.; Rajia, S.; Ogawa, Y.; Fujii, Y.; Yamada, M.; Ozeki, Y.; Kawsar, M.A. Methyl β-D-galactopyranoside esters as potential inhibitors for SARS-CoV-2 protease enzyme: synthesis, antimicrobial, PASS, molecular docking, molecular dynamics simulations and quantum computations. Glycoconjugate J. 2021, 38, 1–30.
- Islam, S.; Hosen, M.A.; Ahmad, S.; Qamar, M.T.; Dey, S.; Hasan, I.; Fujii, Y.; Ozeki, Y.; Kawsar, S.M.A. Synthesis, antimicrobial, anticancer activities, PASS prediction, molecular docking, molecular dynamics and pharmacokinetic studies of designed methyl α-D-glucopyranoside J. Mol. Struct. 2022, 1260C, 132761.
- Kawsar, M.A.; Hosen, M.A.; El Bakri, Y.; Ahmad, S.; Sopi T.A.; Goumri-Said, S. In silico approach for potential antimicrobial agents through antiviral, molecular docking, molecular dynamics, pharmacokinetic and bioactivity predictions of galactopyranoside derivatives. Arab J. Basic Appl. Sci. 2022, 29, 99-112.
Moreover, we found that our compounds interacted with the key residues of these target receptors. Based on these features, we have chosen these targets.
All proteins are taken from those organisms as a test in the biological test.
4QDI- E.coli (expression system)
5A5E- E.coli (Organism and expression system)
7D27- E.coli (expression system)
1ZJI- E.coli (expression system)
3K8E- E.coli (Organism)
2MRW- B.subtilis (Organism)
1EA1- E.coli (expression system)
1AI9- E.coli (expression system)
Comment-6: The size and the center of the docking grid box should be provided.
Response: Correction has been done (section 72.) as below:
“To cover the active site along with the essential residues within the binding pocket, the three-dimensional grid box for docking simulation in which the box with the size of 45.338 x 66.087 x 55.237 was centred using the following dimension, -1.513 x 3.784 x 6.897 used.”
Comment-7: The authors state ''The RMSD and the RMSF the plots of the systems are presented in Figure 10'', but no graph corresponding to the RMSF trajectory has been provided. Please add the RMSF plot.
Response: Correction has been done. We have added the RMSF plot in Figure 10 and also added the related text in the “Abstract, Results and Conclusion.”
Comment-8: Please expand the molecular dynamics section by studying other parameters such as Radius of gyration and Solvent accessible surface area (SASA).
Response: Correction has been done. We have added the Radius of gyration and Solvent Accessible Surface Area (SASA) in Figure 10 and also added the related text in “Abstract, Results and Conclusion.”
Comment-9: The authors should discuss the results obtained from the docking / MD simulations with previous studies.
Response: Correction has been done. We discussed the results obtained from the docking / MD simulations with previous studies.
Reviewer 3 Report
Faez Ahmmed at al. present the synthesis and evaluation of galactopyranoside derivates as antimicrobial agents. I have found several issues in the manuscript which need to be addressed in order for the research to become relevant:
Minor, presentation-related points:
1. The representation of scheme 1 can be improved by adding arrows, indicating the direction of the reaction. Additionally, angles like the reaction from compound 1 to 8 should be avoided to make the figure easier to interpret.
2. The characterization, including listing all NMR shifts and IR bands are part of the method section and not the main manuscript. I would recommend moving these sections to either the method section or the supplementary material.
3. Figure 1 is obsolete since all information presented is already found in the table of Scheme 1.
4. Vegetables are not a good energy source for humans (carbohydrates in form of cellulose are not usable as energy sources for humans, a distinction should be made).
Major, conceptual points:
Spectral information should be given for ALL compounds used in experimental studies. Very critical is that the compounds are indeed clean, pure compounds. Judging from Figure S1 I highly doubt that this is the case. The carbon spectrum shows at least 17 signals, while there are 12 expected signals for compound 2. Similarly, the 1H NMR spectrum looks like one of a non-clean compound as well. With the presented data, I have severe doubts that the compounds used in this study are single compounds, and thereby all following experimental results cannot be attributed to a single compound. This issue needs to be addressed in order to accept any other findings
Author Response
Response to reviewer comments on Molecules 1978757
[N.B.: All corrections have been made by the blue color in the text and highlighted with yellow color]
Response to reviewer- 3 comments
Comment-1: The representation of scheme 1 can be improved by adding arrows, indicating the direction of the reaction. Additionally, angles like the reaction from compound 1 to 8 should be avoided to make the figure easier to interpret.
Response: Thank you so much! Corrections have been done.
Comment-2: The characterization, including listing all NMR shifts and IR bands are part of the method section and not the main manuscript. I would recommend moving these sections to either the method section or the supplementary material.
Response: According to the reviewer’s suggestion, we moved to the supplementary material and also Table-1 and Figure-2 moved into the supplementary section as Table S1 and Figure S3.
Comment-3: Figure 1 is obsolete since all information presented is already found in the table of Scheme 1.
Response: We have moved Figure 1 into the supplementary section as Figure S1.
Comment-4: Vegetables are not a good energy source for humans (carbohydrates in form of cellulose are not usable as energy sources for humans, a distinction should be made).
Response: Correction has been done.
Comment-5: Spectral information should be given for ALL compounds used in experimental studies. Very critical is that the compounds are indeed clean, pure compounds. Judging from Figure S1 I highly doubt that this is the case. The carbon spectrum shows at least 17 signals, while there are 12 expected signals for compound 2. Similarly, the 1H NMR spectrum looks like one of a non-clean compound as well. With the presented data, I have severe doubts that the compounds used in this study are single compounds, and thereby all following experimental results cannot be attributed to a single compound. This issue needs to be addressed in order to accept any other findings
Response: The molecular formula of the compound 2 is C14H23O7Br. So, we expected 14 signals and we mentioned the signals in the spectrum Fig S2 (previously was Fig S1). Moreover, newly we have taken 1H-NMR spectra for clarity and added to the SI.
Thank you very much for your kind support and give valuable time to review our manuscript.
Round 2
Reviewer 2 Report
This paper can be published in the current form
Reviewer 3 Report
While I appreciate the effort to address the issues with the manuscript, my main concern has not been lifted properly.
To interpret any data based on compounds, it has to be shown without doubt that every compound analyzed is pure. This is achieved by the following means:
UV/MS spectrum proving purity of the compound
NMR spectra which shows only expected peaks of the compound. any additional peaks should be interpreted as impurities unless they can be explained.
These two analytical datasets are obligatory for every single compound presented. Otherwise, none of the results can be attributed to the described compound alone.
I would be willing to accept the manuscript if unambigous data is presented that proves the purity of every compound
Author Response
Response: According to the reviewer’s comments, we have added MS and NMR spectra, which were incorporated in the Supplementary Information section (Figure S3 to Figure S19).
Thank you very much.